# MBW: Multi-view Bootstrapping in the Wild

**Mosam Dabhi**[1]      **Chaoyang Wang**[1]      **Tim Clifford**[2]      **László A. Jeni**[1*]

**Ian Fasel**[2*]               **Simon Lucey**[3*]

[1] **Carnegie Mellon University**      [2] **Apple**      [3] **The University of Adelaide**

## Abstract

Labeling articulated objects in unconstrained settings has a wide variety of applications including entertainment, neuroscience, psychology, ethology, and many fields of medicine. Large offline labeled datasets do not exist for all but the most common articulated object categories (e.g., humans). Hand labeling these landmarks within a video sequence is a laborious task. Learned landmark detectors can help, but can be error-prone when trained from only a few examples. Multi-camera systems that train fine-grained detectors have shown significant promise in detecting such errors, allowing for self-supervised solutions that only need a small percentage of the video sequence to be hand-labeled. The approach, however, is based on calibrated cameras and rigid geometry, making it expensive, difficult to manage, and impractical in real-world scenarios. In this paper, we address these bottlenecks by combining a non-rigid 3D neural prior with deep flow to obtain high-fidelity landmark estimates from videos with only two or three uncalibrated, handheld cameras. With just a few annotations (representing 1-2% of the frames), we are able to produce 2D results comparable to state-of-the-art fully supervised methods, along with 3D reconstructions that are impossible with other existing approaches. Our Multi-view Bootstrapping in the Wild (MBW) approach demonstrates impressive results on standard human datasets, as well as tigers, cheetahs, fish, colobus monkeys, chimpanzees, and flamingos from videos captured casually in a zoo. We release the codebase for MBW as well as this challenging zoo dataset consisting of image frames of tail-end distribution categories with their corresponding 2D and 3D labels generated from minimal human intervention.

## 1   Introduction

Hand labeling landmarks of articulated objects within video is an arduous and expensive task. Landmark detectors [29, 20, 32] can be employed to automate the process. However, they require the ingestion of large amounts of labeled training data to be reliable – an infeasible requirement for all but the most common of articulated objects (e.g. people, hands). Semi-supervision can help [28], where a small portion of frames within the video are hand labeled. Candidate labels can be generated from the noisy landmark detectors – trained from the seed hand labeled examples – inliers are then determined through calibrated rigid multi-view geometry. These inliers are treated as labels and used to train the next round of landmark detectors. This semi-supervised process is iterated to increase the number of inlier estimates, with additional human annotation being added judiciously to ensure the full sequence is labeled. Such strategies have been instrumental for obtaining reliable ground-truth – most notably the Multi-view Bootstrapping (MB) approach of Simon et al. [28]. Human annotators

---

*indicates the authors advised equally

36th Conference on Neural Information Processing Systems (NeurIPS 2022) Track on Datasets and Benchmarks.

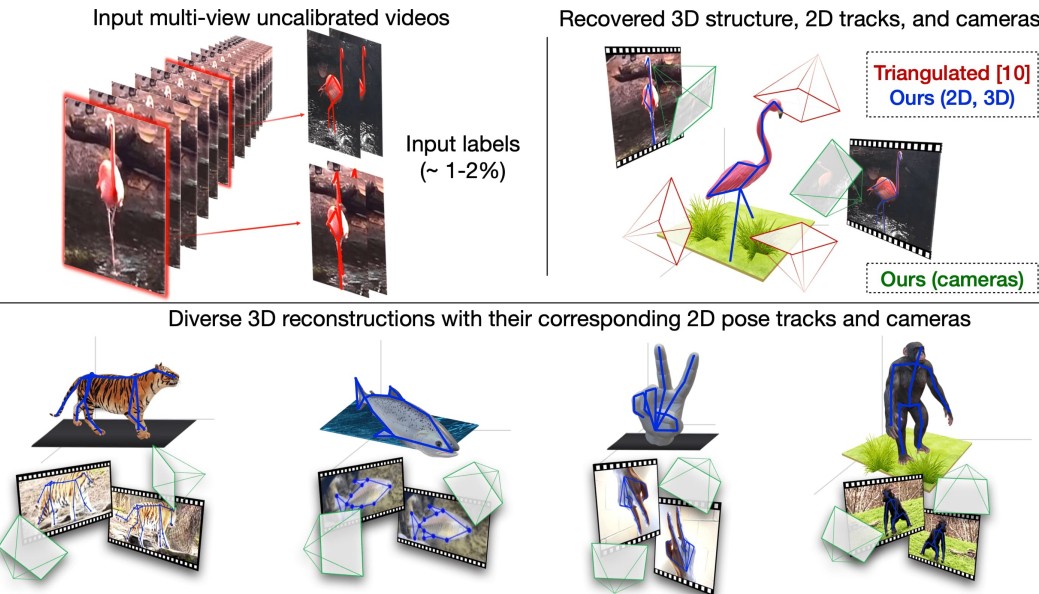

Input multi-view uncalibrated videos

Input labels (~ 1-2%)

Recovered 3D structure, 2D tracks, and cameras

Triangulated [10]
Ours (2D, 3D)

Ours (cameras)

Diverse 3D reconstructions with their corresponding 2D pose tracks and cameras

Figure 1: Overview of our MBW approach. **Top**: Provided an unconstrained Multi-view uncalibrated video with very few 2D labels ($\approx 1-2$ % or $\approx 15$ labels), our method recovers the 3D structure in a canonical frame, along with camera poses and corresponding 2D landmarks for the complete video sequence. **Bottom**: Diverse reconstructions and data labeling for videos captured in the wild. This dataset is released as part of the paper.

are only required to hand label a subset of the dataset, with the rest just requiring visual inspection to validate the accuracy of the inferred labels.

Although significantly cutting down on human labor, Multi-view Bootstrapping [28] is still expensive and cumbersome, requiring a static multi-camera rig which usually consists of tens [1] or even sometimes hundreds of calibrated cameras [11]. The number of cameras can be reduced, but with a trade-off in decreasing robustness of outlier rejection and increasing human interventions (see Fig. 4). This makes it less feasible for capturing objects outside laboratories. In this paper, we advocate for a significant advancement by enabling its application to data captured by a few (2 to 4) handheld cameras with only a handful of annotated frames (about 10-15 frames per several minutes of video). We refer to our approach herein as *Multi-view Bootstrapping in the Wild* (MBW). The cameras need not be calibrated, and fields of view need only overlap the articulated object, not the backgrounds.

Our innovations come from (i) utilizing Multi-View Non-Rigid Structure from Motion (MV-NRSfM) [2] to more reliably estimate camera poses and 3D landmark positions from noisy 2D inputs with few cameras. Compared to performing SfM / triangulation independently for each frame as in prior works [11, 1], MV-NRSfM leverages the redundancy in shape variations among different frames, thus it is less sensitive to the variations of input views, more capable of detecting outliers and denoising inlier 2D landmark estimates. (ii) We leverage recent advances in deep optical flow [30] as an alternative strategy for creating landmark label candidates – something especially useful in the early iterations of the semi-supervision process.

As a result our approach can be effectively applied to less studied articulated object categories. We show results on tigers, fish, colobus monkeys, gorillas, chimpanzees, and flamingos from a zoo dataset (captured by the authors, who hereby release it under a CC-BY-NC license). We also quantitatively evaluate the proposed pipeline on common motion capture datasets (*e.g.* Human3.6 Million [9]). The accuracy of the learned landmark detector is competitive to state-of-the-art fully supervised method. A graphical depiction of our approach can be found in Figure 1.

## 2 Related Works

Panoptic Studio [11] paved the way for collecting data for deformable objects such as the human body. Subsequent efforts on humans [9, 24, 36], hands [38, 39, 19], monkeys [1], canines [14], chee-

Table 1: Related efforts trying to achieve a similar application as the proposed approach.

| Method | Flow | Calibration | 3D labels | Wild setup | % annotated ($\approx$) |
|---|---|---|---|---|---|
| Günel et al. [5] | No | Required | **Yes** | No | 30% |
| Mathis et al. [18] | No | N/A | No | No | 5% |
| Dong et al. [3] | **Yes** | Required | No | No | N/A (Unknown) |
| Zhang and Park [37] | **Yes** | Required | No | No | 4% |
| Pereira et al. [22] | No | N/A | No | No | 5% |
| Simon et al. [28] | No | Required | **Yes** | No | 30% |
| **MBW (Ours)** | **Yes** | **No** | **Yes** | **Yes** | **2**% |

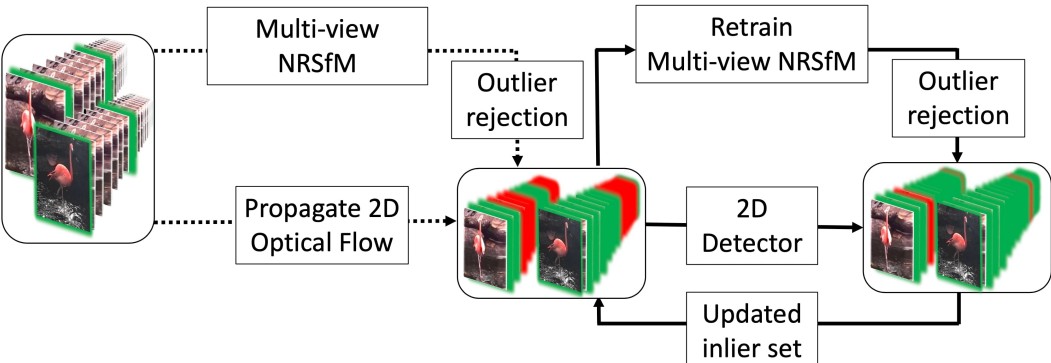

Figure 2: (Dotted lines) The MV-NRSfM neural shape prior is initially trained with labels for 1-2% of the frames (shown as green images). A pre-trained optical flow network then propagates the initial labels through the video to generate additional 2D candidates. Candidates that result in high reprojection error from the 3D lifting network are rejected as outliers (red). (Solid line) From here on, the label set is updated with inliers from the previous iteration, and is then used both to retrain the MV-NRSfM and to train a 2D detector. Dotted line is executed only once while solid lines are repeated for $K$ iterations.

tahs [12], rats [16], and insects [5] have followed. Multi-view Boostrapping [28] has demonstrated how these calibrated multi-camera datasets can be labeled efficiently through a semi-supervised learning paradigm and a small number of hand annotations. A fundamental drawback to multi-view bootstrapping however is that it requires a large number of views and accurate camera calibration.

Recent works have explored alternate paradigms for semi-supervised landmark labeling that do not require such exotic calibrated multi-camera setups. Mathis et al. [18], Pereira et al. [22], and Yu et al. [35] tackle this problem from a single view, but largely ignore the use of multi-view geometry. Gunel et al. [5] have explored an approach that utilizes a small number of camera views, and only requires an approximate estimate of the camera extrinsics. They use pictorial structures [4] to automatically detect and correct labeling errors, and use active learning to iteratively improve landmark detection performance. Although this approach is useful in lab settings where there are static cameras and the object is anchored to a fixed location (e.g. tethered flies are positioned over a spherical treadmill [5]), it is non-trivial to generalize such performance to more complex environments and across significant individual variations due to e.g. patterned skins in animals or demographics and clothing in humans. In contrast, our approach accepts image frames from moving cameras and requires only a handful of hand annotated labels. Further, it does not require any camera information, and can easily be applied to a broad set of articulated objects such as humans, hands, and animals. Thus, the strength of our method is its generalizability. Since the provided implementation of DeepFly3D was specific for Drosophila, it was not readily applicable to our in-the-wild datasets. An overview highlighting major differences between our proposed approach and related works trying to achieve a similar application is shown in Tab. 1.

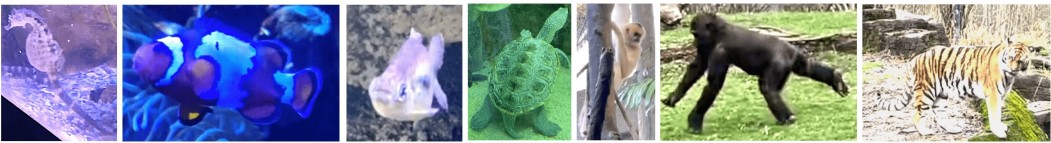

Figure 3: Sample sequences composited from our Zoo data collection – situations where traditional SLAM pipelines fail to recover reasonable camera matrices due to lack of reliable matching features.

## 3 Approach

### 3.1 Problem Setup

Our goal is to learn 2D landmarks of articulated objects from multi-view synchronized videos captured in the wild. Unlike other works [11, 5, 3, 37] developed for laboratory settings, we focus on the *in the wild* setting, *i.e.* data is captured using a small number (2 or 3) of cameras with *unknown* extrinsics, and only a small portion (1 to 2%) of the data is manually labeled.

More specifically, our training set $\mathcal{S}$ consists of $V$ synchronized videos, each with $N$ frames. Each training image is denoted as $\mathbf{I}_{(n,v)}$ where $n \in [1, \dots, N]$ and $v \in [1, \dots, V]$ denote frame and view indices. Initially only a subset of frames $(n, v) \in \mathcal{S}_0$ are given with 2D landmark annotations $\mathbf{W}_{(n,v)} \in \mathbb{R}^{P \times 2}$ of $P$ points. Each row of $\mathbf{W}_{(n,v)}$ corresponds to the 2D location of a landmark (*e.g.* the left knee of flamingo, see Fig. 3). To simplify explanations, we assume that only a single object of interest is visible in each frame. For multiple non-overlapping objects, our algorithm is able to estimate bounding boxes to reduce the problem into a single object case (see Appendix D). Finally, the goal is to (i) infer the missing 2D landmark annotations in the training set as a self-labeling task; (ii) train a 2D landmark detector for unseen objects of the same category.

### 3.2 Learnable geometric supervised self-training

We employ a self-training approach which iteratively assigns pseudo labels and retrains a 2D landmark detector. At each iteration, the 2D pseudo labels generated by a landmark detector are verified using geometric constraints. Samples which fail the verification are dropped, and the remaining pseudo labels are denoised before feeding them back as labels to retrain the landmark detector. Such geometric supervised self-training strategy has been widely used in learning landmark detections [5, 11, 1, 28], what differentiates our work is that we model the geometric constraints as a *learnable* function, which is learned together with the landmark detector. We abstract this function as:

$$g : \tilde{\mathbf{W}}_1, \tilde{\mathbf{W}}_2, \dots, \tilde{\mathbf{W}}_V \longrightarrow y_1, y_2, \dots, y_V \tag{1}$$

where $\tilde{\mathbf{W}}_v \in \mathbb{R}^{P \times 2}$ represents detected 2D landmarks at $v$-th view, and $y_v$ is the measured uncertainty for outlier rejection. We derive $g$ from performing multi-view non-rigid structure from motion (MV-NRSfM) as described in Sec. 3.3. The remaining details of the self-training pipeline is given as follows.

**Initialization** In the initial step, we require human labelers to annotate the 2D landmark positions of the same target object for a small portion of captured video frames. We then train our geometric constraint function $g$ using these initial labels. Since the initial labels only cover a limited range of shape variations, the learned $g$ is aggressive in detecting outliers at the beginning stage of the training. It will be improved as it sees more shape variation in each iteration.

**Label propagation through tracking** We find that directly training a 2D landmark detector such as HRNet [29] using very few labeled samples yields unstable results. To increase the number of training samples, we propagate the annotated 2D landmark labels to the rest of unlabeled frames through tracking. We use an off-the-shelf optical flow network [30] to track the landmarks frame to frame. Other tracking methods [25, 6] can also be utilized. We employ standard forward&backward flow consistency check to detect tracking failures. Since the optical flow network tends to make consistent wrong estimations when swapping the input frames, such consistency check alone is not enough to exclude all tracking failures. Therefore, we further employ the learned geometric constraint function $g$ to aggressively remove any likely outliers if the predicted uncertainty $y$ is above a certain threshold. We then add the remaining tracked points (inliers) to the labeled set. This new set is

then used both to re-train $g$, and to train the first iteration of the 2D landmark detector used in the subsequent stages.

**Self-training iterations** At each iteration $t$, we define a "labeled" set $\mathcal{S}_{t-1}$ which includes all frames that are either manually annotated, or are labeled by the landmark detector $f_{t-1}$ in the previous stage and passes outlier rejection using $g_{t-1}$. We then re-train the landmark detector and the geometric constraint function on the labeled set $\mathcal{S}_{t-1}$, which leads to a new detector $f_t$ as well as $g_t$. Once trained, inference is run with this detector network $f_t$ over all the captured frames. This produces new pseudo labels $\tilde{\mathbf{W}}_{n,v}^t$ for all the $N$ frames and $V$ views. We then apply the geometric constraint function $g_t$ to evaluate the uncertainty score $y_{n,v}^t$ for each pseudo label. Finally we define a new labeled set $\mathcal{S}_t$ which includes all samples $(n,v)$ that satisfy $y_{n,v}^t$ is below a certain threshold.

The above process is repeated for a number of iterations. In principle frames that are still not annotated (rejected by $g_t$) can be actively labeled by humans, however in practice we have found this situation is rare, unless the distance between the captured views is extremely small, making it difficult to learn a reasonable 3D shape prior.

### 3.3 Outlier detection using multi-view NRSfM network

**Uncertainty score.** Our geometric constraint function $g$ is built upon measuring the discrepancy between detected 2D landmarks and the 3D reconstruction by a multi-view NRSfM method. This is in the same spirit as using the reprojection error of triangulation to measure uncertainties as in prior works. The idea is if the detected 2D landmarks at different views are all correct, we should be able to recover accurate camera poses and 3D structures, and consequently the reprojection of recovered 3D landmarks matches the 2D landmarks. On the other hand, if the reprojection error is high, it means there exists errors in the 2D landmarks which prevents perfect 3D reconstructions. This leads to the following formulation of our uncertainty score,

$$y_{(n,v)} = \|\tilde{\mathbf{W}}_{(n,v)} - \mathrm{proj}(\tilde{\mathbf{T}}_{(n,v)}\tilde{\mathbf{S}}_n)\|_F \tag{2}$$

where $\tilde{\mathbf{T}}_{(n,v)}$, $\tilde{\mathbf{S}}_n$ are the estimated camera extrinsics and 3D landmark positions in the world coordinate, $\tilde{\mathbf{W}}_{(n,v)}$ is 2D landmarks estimated by the landmark detector, and proj is the projection function. The effectiveness of the uncertainty score defined by Eq. 2 depends on the reliability of estimating $\tilde{\mathbf{T}}_{(n,v)}$, $\tilde{\mathbf{S}}_n$. However, due to the low number of synchronized views as well as noise in $\tilde{\mathbf{W}}_{(n,v)}$, simply performing SfM and triangulation gives poor result as shown in Fig. 4a. This motivates the following use of MV-NRSfM.

**Unsupervised learned MV-NRSfM.** Our solution to reliably estimate $\tilde{\mathbf{T}}_{(n,v)}$, $\tilde{\mathbf{S}}_n$ is to marry both the multi-view geometric constraints and the temporal redundancies across frames, which leads to the adaption of the MV-NRSfM method [2]. Limited by space, we refer interested reader to their paper for detailed treatment. Here we briefly discuss its usage in our problem. In a nutshell, MV-NRSfM [2] assumes that 3D shapes (concatenation of 3D landmark positions) can be compressed into low-dimensional latent codes if they are properly aligned to a canonical view. MV-NRSfM is then trained to learn a decoder $h_d : \boldsymbol{\varphi} \in \mathbb{R}^K \to \mathbf{S} \in \mathbb{R}^{P \times 3}$ which maps a low-dimensional code to an aligned 3D shape, as well as an encoder network $h_e : \mathbf{W}_1, \mathbf{W}_2, ..., \mathbf{W}_V \to \boldsymbol{\varphi}$ which estimates a single shape code $\boldsymbol{\varphi}$ from 2D landmarks $\mathbf{W}_v \in \mathbb{R}^{P \times 2}$ captured from a number of different views (see Appendix C for the network architecture). Both $h_d$ and $h_e$ are learned through minimizing the reprojection error:

$$\min_{\mathbf{T}_{(n,v)}, h_d, h_e} \sum_{(n,v) \in \mathcal{S}} \|\tilde{\mathbf{W}}_{(n,v)} - \mathrm{proj}(\mathbf{T}_{(n,v)}(h_d \circ h_e)(\tilde{\mathbf{W}}_{(n,1)}, \tilde{\mathbf{W}}_{(n,2)}, ..., \tilde{\mathbf{W}}_{(n,V)}))\|_F \tag{3}$$

where $\mathcal{S}$ refers to the training set, and $\circ$ denotes function composition. Thanks to the constraint from low dimensional codes as well as the convolution structure of $h_e$ inspired from factorization-based NRSfM methods [15], the learned networks $h_d \circ h_e$ are able to infer reasonable 3D landmark positions from noisy 2D landmark inputs. We provide the network architecture of MV-NRSfM in Appendix C.

In our task, we rely on the robustness of MV-NRSfM not only to learn the 3D reconstruction of the labeled training set, but also to detect outliers on the unlabeled set using Eq. 2. At the $t$-th iteration of our self-training, we train $h_d^t$, $h_e^t$ given the current labeled set $\mathcal{S}_{t-1}$ from the previous iteration. We then test $h_d^t \circ h_e^t$ on the detected 2D landmarks from the unlabeled set to produce $\tilde{\mathbf{S}}_n$ used in Eq. 2.

Camera extrinsics $\tilde{\mathbf{T}}_{(n,v)}$ are then estimated simply through either an orthographic-N-point (OnP) or perspective-N-point (PnP) solver depending on the choice of camera projection model. In our data, we find that assuming a weak perspective camera and use OnP already gives high fidelity results.

Finally, We note that the unsupervised learned MV-NRSfM networks *i.e.* $h_d \circ h_e$ is likely not able to estimate correct 3D landmarks if its 2D inputs are significantly different than its training set. Instead, it tends to output a plausible 3D structure but does not fully match the 2D inputs. This is actually a desirable behavior for our task, since it serves the purpose of out-of-distribution (OOD) detection – detecting any shapes that differ significantly to the current labeled set. We expect the MV-NRSfM to cover full shape variations in the input sequences as the "labeled" set expands while the training progresses and give a detailed analysis in Appendix A.

# 4 Experiments

Our experiments aim to answer the following questions: **(I)** Is MBW able to generate reliable 2D and 3D landmark predictions from limited views (as few as two) given only a few (as few as 1-2%) human labels? **(II)**: Is MBW able to reject outliers and learn a meaningful shape distribution from these few input labels? **(III)** How important is the number of views in our pipeline? **(IV)** Can MBW refine (denoise) the 2D candidate inliers? **(V)** Is our pipeline able to compete with leading benchmarks despite using a fraction of input 2D labels? Before diving into our experiments, we discuss the details of our pipeline.

**Datasets.** Datasets with multi-view videos of non-human subjects are rare, so we collect our own dataset of animals. The collected **zoo dataset** consists of five animal categories, each with 2 synchronized videos. The videos contain viewpoint and dynamic appearance changes as well as common imaging artifacts such as reflection of water or blurred frames (see Fig. 3). For this data we manually annotated part of the sequences for evaluation. In addition, we used the benchmark dataset of Human3.6M [9] (H36M) to perform quantitative evaluations of our approach.

**Implementation Details** We train our approach on 1 NVIDIA RTX 3090 GPU with 24 GB memory. A learning rate of 0.001 is used for all networks. We train each network from scratch. A pre-trained RAFT network [30] is used with flow iterations of 20. Bottleneck size of 8 is used for MV-NRSfM [2] for all categories. We use HRNet [29] as the backbone 2D detector, and the same configuration is used for all object categories.

**Question I: Limited amount of labels and views** We use just two camera views from Directions-1 sequence of Subject #1 from H36M dataset [9]. Each camera view consists of 1383 frames per-view, amounting to 2766 frames in total. Of these, we provide hand labels for only 20 frames (10 frames per view amounting to 0.8% of the total frames) through uniform sampling. Our task is to generate 2D landmark predictions of the remaining frames of this sequence ( 99.2% unlabeled).

We evaluate the accuracy of 2D landmark predictions using the commonly used evaluation metric of PCK by Andriluka et al. [33]. We report area under the curve of PCK at different thresholds to understand the nature of 2D prediction errors over all the frames. For consistency, the 2D landmark error is normalized using head bone length before evaluation. As baseline, we keep the complete architecture of MBW, but replace MV-NRSfM with multi-view triangulation using groundtruth calibrated cameras to reject outliers and denoise inliers [28]. We denote this baseline as Trng.

2D landmark prediction performance over all the frames is shown in Fig. 4c. The quantitative results are shown in Tab. 2 where we we report PCK AUC values to evaluate 2D landmark prediction accuracy. We evaluate the 3D structure accuracy using Procrustes-Aligned Mean Per Joint Position Error (PA-MPJPE) [27]. This metric evaluates 3D joint localization accuracy in mm and represents the $L2$ distance between the groundtruth and predicted joint locations after aligning the 3D structures using a rigid transformation. Table 2 shows that our approach is able to generate high-fidelity 2D landmark prediction as well as accurate 3D structure despite starting from a mere 0.8% of 2-View data. In contrast, the competing baseline fails since it cannot reconstruct good 3D structure from just 2 views and extremely noisy landmark predictions. This experiment helps us answer Question **(1)**: **Yes**, MBW with MV-NRSfM is able to predict reasonable 2D and 3D landmark prediction using small amount of labels and views.

**Question II: Outlier rejection** The proposed pipeline requires bad 2D landmark candidates (outliers) to be rejected so they are not incorporated into subsequent training iterations as (inlier)

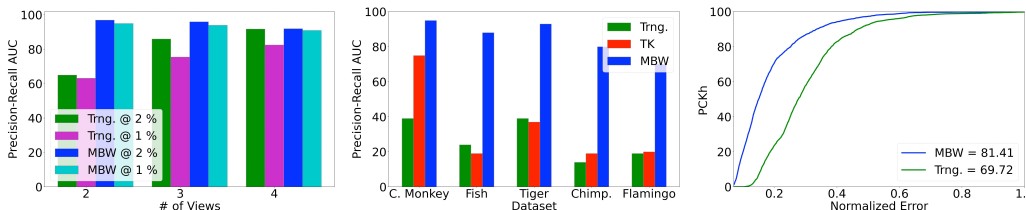

|                        |                          |                          |
| :--------------------: | :----------------------: | :----------------------: |
| (a) Outlier detection on [9]. | (b) Outlier detection on Zoo dataset | (c) 2D accuracy using PCKh [33]. |

Figure 4: (a) Precision-Recall (PR) AUC shows the outlier rejection accuracy on benchmark [9] with varying number of camera views and input 2D labels. PR AUC bars per each view are calculated using the frames available in that view. Fewer views corresponds to a smaller number of frames, and more views comprise a larger number of frames. (b) Outlier rejection accuracy using PR AUC on zoo dataset. (c) Final 2D landmark prediction accuracy using our approach compared to multi-view triangulation (with groundtruth cameras) [8] (Trng). We calculate the normalized error (dividing by head bone length [33]) and report the PCK AUC result varying the error threshold range. Note that we run this experiment on just 10 frames per view (amounting to $0.8\%$) and only two views.

Table 2: Quantitative comparison showing improvement in 2D detection compared to comparative baselines (with groundtruth camera) on benchmark dataset [9].

| Method | 2D ↑ | 3D (mm) ↓ |
| :--- | :---: | :---: |
| Triangulation [8] | 69.72 | 1383 |
| MBW (Ours) | **81.41** | **39.3** |

pseudo-labels $\mathcal{S}_i$. To evaluate this, we design the following experiment to evaluate the Precision-Recall AUC on our **zoo** dataset as shown in Fig. 4b. The number of initial 2D manual labels provided to each object is $2\%$. We perform an ablation study in which we evaluate at the initial iteration of MBW, to check MBW's ability to reject and refine noisy landmark predictions from optical flow.

MBW is compared against weak-perspective Tomasi-Kanade Structure-from-Motion (TK) [31] – the only other approach able to reconstruct 3D structure without calibrated cameras. For completeness, we also estimate cameras in the wild using [26] so that we can compare against Trng [7]. For the competing approaches, we use our whole pipeline as-is except for the MV-NRSfM outlier rejection and denoising part, instead using the competing method. Since there is no groundtruth in the **zoo** dataset, we manually label each 2D flow prediction and assign inlier labels by visual verification. The PR AUC is calculated using this manually generated groundtruth verification. As evident in Fig. 4b, only our approach can accurately reject the outliers as evident by high PR AUC. This finding is reasonable since we found that [26] is unable to calculate cameras due to non-existent or poor matching features across views collected in the **zoo** dataset, resulting in no 3D reconstruction. Another baseline, weak-perspective Tomasi Kanade SfM is unable to handle the noise encountered in real-world data and hence fails to accurately reconstruct 3D. Thus, the above experiment helps us answer Question (**II**): **Yes**, MBW is able to reject outliers despite of having limited input 2D labels, number of views, and no camera information.

**Question III: Number of Views** For this experiment, we take the same H36M dataset as noted in Question 1, with the exception that we vary the number of views (2-4) and number of 2D input labels (1-2%) provided to our approach and competing baseline of Trng. In Fig. 4a we see that both our approach and the baseline reject outliers reasonably well for four views. However as we reduce the number of views Trng. had a steeper dropoff due to unreliable geometric constraints arising from limited views and noisy 2D labels. In contrast, since MBW uses a learned shape prior for rejecting outliers, our performance remains consistent across varying views. This experiment helps us answer Question (**III**): **Yes**, as long as we learn a good shape prior, we do not require large number of views to reliably reject the outliers.

**Question IV: Denoising capability** To answer this question, we report the visual results of the 2D landmark predictions by MBW on **zoo** dataset in Fig. 5. On the left, we see that MBW is able to detect bad 2D predictions without human intervention (also evident by PR AUC in Fig. 4b). On the right, we highlight scenarios where our approach can use the MV-NRSfM shape prior information to

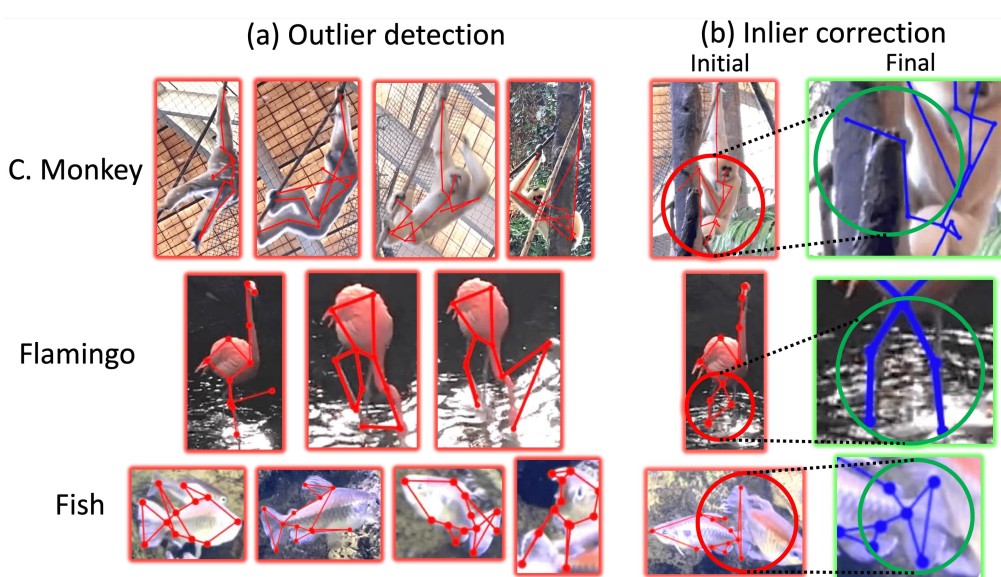

Figure 5: (a) Shows example outliers detected using the proposed approach. (b) Shows the denoising capability of the proposed approach where we are able to denoise the improve the inliers by using the multi-view shape prior information.

Table 3: 3D pose reconstruction accuracy of different methods on the Human3.6M dataset [9]. Our approach uses just 2% of 2D labels yet achieves 3D performance comparable to competing methods.

| Multi-view methods | (mm) ↓ |
|---|---|
| Martinez et al. (multi-view). [17] | 46.5 |
| Pavlakos et al. [21] | 41.2 |
| Kadkhodamohammadi & Padoy [13] | 39.4 |
| Iskakov et al. [10] | 19.9 |
| Reddy et al. [23] | **17.5** |
| Ours (PA-MPJPE at 2%) | 20.15 |

not only detect but refine (denoise) the inliers. Moreover, apart from qualitative visual inspection, we also conduct a toy study to analyze the denoising capabilities of our approach and discuss this in the supplementary section. The visual inspection experiment in Fig. 5 helps us answer Question (**IV**): **Yes**, MBW can effectively denoise candidate inliers.

**Question V: Comparison against leading benchmarks**    Finally, we compare the performance of our approach with existing state-of-the-art semi-supervised learning approaches. Since our approach has overlap with MSBR [37] in that they have an initial labeled set, as well as an unlabeled set, we compare and report the 2D landmark prediction accuracy using the same experimental setup as theirs and report in Tab. 4. To reiterate, similar to them we use Eating and Discussion but just use 2% as input labeled data and test on all frames of Greeting for a fair comparison. We observe that in spite of using a small amount of 2D input labels, our approach is able to compete against well-established benchmark approaches that use >30% of frames to label – making our approach more efficient in terms of effort required to generate labels of similar accuracy.

Lastly, we conduct 3D reconstruction analysis on the Directions-1 sequence of H36M [9] using the common protocol where Subjects 1,5,6,7,8 are used as train sets, and S9, and 11 are used as the test set. We just provide 2% of train set labels as 2D input labels to our approach and evaluate the 3D reconstruction performance on 100% of the test set, and report the 3D reconstruction results in Tab. 3. Since our approach calculates 3D reconstruction in a canonical frame and is up-to-scale, we apply the rigid transformation of Procrustes Alignment and report the 3D landmark reconstruction accuracy as PA-MPJPE in mm. We observe that we are able to compete with state-of-the-art benchmarks despite using 2% input labels and no camera information in our approach. The above two experiments helps us answer Question (**V**).

Table 4: We compare our approach with existing semi-supervised learning frameworks: (1) temporal supervision [3] and (2) cross-view supervision [34, 37]. We evaluate on human dataset (Human3.6M [9]) using PCKh measure. We test the generalizability by applying on unseen data. We use just **2**% of the labeled data compared to other approaches.

| Human3.6M [9] | Nec ↑ | Sho ↑ | Elb ↑ | Wri ↑ | Hip ↑ | Kne ↑ | Ank ↑ |
|---|---|---|---|---|---|---|---|
| Dong et al. [3] | 91.7 | 81.4 | 42.3 | 25.6 | 93.9 | 83.4 | 87.5 |
| Jafarian et al. [34] | 89.6 | 48.3 | 29.7 | 20.5 | 29.8 | 34.9 | 60.7 |
| Zheng & Park [37] | 93.2 | **92.8** | 67.3 | 49.6 | **93.7** | **87.6** | 89.5 |
| MBW (Ours) | **96.8** | 83.3 | **78.1** | **69.8** | 89.2 | 82.9 | **92.9** |

## 5 Limitations

Our approach relies on a 3D neural shape prior to address challenging scenarios such as occluded or noisy keypoints. However, it is only able make accurate predictions if the views have sufficiently wide baseline, for instance to address cases where an occluded keypoint in one view is unoccluded in another view. In these scenarios MBW can reject (or clean) the occluded keypoint through the 3D prior. However if the keypoint is occluded in both (all) the available views, then MBW fails to produce meaningful predictions. This is expected since MBW cannot hallucinate without any cues or priors. Some more recent works such as particle trajectory tracking [6] may help address the issues of occlusion, and we plan to take these lines of work into account in future work.

## 6 Conclusion

The idea of leveraging geometry to make labeling easier is not new – indeed this is the basis for large-scale dome-based data collection methods [11, 1, 19, 39, 14]. Our key contribution is incorporating a neural prior to enable the same ideas to be applied to tail-end distribution non-rigid object categories. This has yielded several key insights: **(a)** Deep 3D neural shape priors trained on even as few as $10 - 15$ frames per video can already provide powerful constraints for modeling deformable objects; **(b)** provided a reliable outlier rejection method, optical flow-based methods [30] provide a simple and effective way to propagate 2D candidate predictions in videos; **(c)** neural priors can be used as the basis of reliable outlier rejection, enabling bootstrapping under constrained and error-prone methods such as flow and "partially trained" 2D networks into highly accurate vision pipelines. Our approach is just a first step in this direction, but it may point the way to a revival of iterative pipelines that more deeply integrate the insights of recent neural approaches with iterative refinement methods familiar in classic 3D computer vision.

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
