# OpenReview forum: "MBW: Multi-view Bootstrapping in the Wild"
_NeurIPS.cc/2022/Track/Datasets_and_Benchmarks — NeurIPS 2022 Datasets and Benchmarks _

### Official Review · Reviewer_XYGa · 2022-07-04
**good paper; design well justified; comprehensive analysis; accept**

**Rating:** 7
**Confidence:** 3
**Clarity:** 1. model design is well motivated.
2.…

**Strengths:**

1. paper is well-written and well-organized with great clarity on design motivations
2. the arguments made in the paper are justified and backed up by experimental results. Comprehensive analysis is provided to better understand the design motivations and model behaviors.
3. the performance improvements in 2D and 3D results are significantly greater than in baselines
4. the idea of active learning is not new; but utilizing multi-view for bootstrapping in the wild is interesting. It could extend to other applications in 3D vision.

**Weaknesses:**

1. it would be great if the authors can conduct experiments to demonstrate the effectiveness of the method in more unconstrained natural environments when there involve large camera motions as well. (*quantify how large camera motions would impact the results)
2. test generalization abilities when context changes (e.g. tigers move from one bridge cage to another dark cage in the zoo) and quantify how the method could quickly adapt to continuously changed domains

**Additional Feedback:**

see weakness section

**Correctness:**

yes, the claims seem to be correct as far as I understand from the experimental results.
The evaluation methods and designs are appropriate and convincing

**Documentation:**

yes, Github link is provided with the dataset link for public use
However,
more details on the camera setup in the zoo settings can be specified in more details in the supplementary material (e.g. the poses and subtle motions of camera setups, statistics on lighting, color, animal instances, background, and so on)
One side question, do the authors plan to expand this zoo dataset further?

**Ethics:**

no ethical concerns as far as I can think of

**Relation To Prior Work:**

yes, the authors have explicitly identified key differences between current approaches and others in the related works. These include temporal consistency, camera calibration, 3D labels, number of annotations and wild setups.

**Summary And Contributions:**

Hand labeling objects in natural videos is challenging and labor-intensive. Instead of using rigid geometry and calibrated cameras, the paper proposed an approach of combining a non-rigid 3D neural prior with the deep optical flow to obtain good-quality landmark detection. The approach requires annotations from 1-2% of the video frames from only 2-3 uncalibrated handheld cameras. The authors conducted a comprehensive analysis in 2D and 3D reconstructions of the remarkable experimental results on two benchmark datasets: human and newly curated datasets.

---

> ### Author Response · Authors · 2022-08-14
> **Response to Reviewer XYGa feedback**
>
> **[Weaknesses 1.] It would be great if the authors can conduct experiments to demonstrate the effectiveness of the method in more unconstrained natural environments when there involve large camera motions as well. (*quantify how large camera motions would impact the results)***
>
> The proposed method should work for most of the casually captured handheld camera videos in the wild as long as the object of interest is visible in at least one of the views. Frames with severe motion blur and sudden changes in lighting may negatively impact the optical flow propagation part of our method and may require human annotation if they are not included in the "inliers" category after multiple iterations of MBW.
>
> &nbsp;
>
> **[Weaknesses 2.] Test generalization abilities when context changes (e.g. tigers move from one bridge cage to another dark cage in the zoo) and quantify how the method could quickly adapt to continuously changed domains}**
>
> Changing context domain is indeed a challenging scenario and our current strategy to handle such cases is to require an initial set of annotations for both domains. As long as the object (tiger) is visible and as long as there exist a few amounts of frames in the changing domains, MBW should be able to generate the annotations in such scenarios.

---

> > ### Comment · Reviewer_XYGa · 2022-08-15
> > **thank you for the response**
> >
> > The authors have clarified my initial doubts and concerns. My rating remains as 7 - accept.

---

### Official Review · Reviewer_QwEG · 2022-07-27
**A scalable semi-supervised pipeline for 3D landmark annotation of articulated objects**

**Rating:** 7
**Confidence:** 4
**Clarity:** Yes. The paper is well written and ea…

**Strengths:**

1. This paper proposes a feasible semi-supervised pipeline for 2D and 3D landmark annotation of articulated objects under in-the-wild setting. In contrast to complex capture systems like Panoptic Studio that can only be used in the laboratory and applied to limited objects, the proposed pipeline is more flexible, and has the potential to be applied to a wide variety of objects. Collecting large-scale dataset is possible using the proposed pipeline, as only sparse manual annotation is necessary.
2. The authors successfully apply the proposed annotation pipeline to the newly collected “zoo dataset”. This shows the effectiveness of the proposed pipeline, while providing researchers with valuable data of less studied articulated object categories.
3. Extensive experiments show the reliability of the proposed annotation pipeline.

**Weaknesses:**

1. This paper focuses on the semi-supervised annotation pipeline rather than the dataset itself. However, the novelty of the proposed pipeline is limited, as it mainly takes advantage of MV-NRSfM and optical flow.
2. Although the proposed pipeline outperforms baseline methods, it is not clear whether the annotation quality is sufficient to be used as ground truth. More analyses of the pipeline may be necessary, e.g., what should be done when the semi-supervised annotation quality is not good enough? Does adding more manual annotation help? If so, how many additional annotations would be meaningful? From Fig 4(a), increasing manual annotation from 1% to 2% does not seem to significantly improve the outlier rejection.
3. There is limited discussion on the “zoo dataset”. It would be nice if the authors can provide more in-depth analyses of the dataset, e.g., what is the potential usage of the “zoo dataset” given it is annotated by the semi-supervised pipeline?

**Additional Feedback:**

1. More analyses on the robustness of the semi-supervised annotation pipeline would be helpful. For example, does it always work when the camera baseline is sufficiently large (all keypoints are visible from at least one view)? If not, what would be the typical failure pattern?
2. The authors can showcase the value of the “zoo dataset” by applying it to some downstream tasks.

**Correctness:**

Yes. The claims made in the submission are correct. The dataset is constructed in a sound way.

**Documentation:**

Yes. There is sufficient detail on data collection and organization, availability and maintenance, and ethical and responsible use.

**Ethics:**

No. There is no ethical concern.

**Relation To Prior Work:**

Yes. The difference of this work from previous works is clearly discussed.

**Summary And Contributions:**

This paper proposes Multi-view Bootstrapping in the Wild (MBW), a semi-supervised pipeline that can annotate 2D and 3D landmarks of articulated objects given videos captured by two or three uncalibrated handheld cameras, where 1-2% of frames are manually annotated. Specifically, the proposed pipeline builds a neural shape prior from sparse manual annotations using Multi-View Non-Rigid Structure from Motion (MV-NRSfM), then propagates the initial labels through the entire video using a pre-trained optical flow method. The propagated 2D landmark candidates are used to iteratively retrain the MV-NRSfM as well as a 2D detector. The authors collect the “zoo dataset” and apply the proposed pipeline for 2D and 3D landmark annotation. Extensive experiments on the “zoo dataset” and the Human3.6M dataset show the reliability and scalability of the proposed annotation pipeline, i.e., given very limited number of camera views and manual annotations, the proposed pipeline can reliably annotate the 2D and 3D landmarks while rejecting (correcting) outliers.

---

> ### Author Response · Authors · 2022-08-14
> **Response to Reviewer QwEG feedback (Weaknesses #2)**
>
> **[Weaknesses 2(a)] Although the proposed pipeline outperforms baseline methods, it is not clear whether the annotation quality is sufficient to be used as ground truth.**
>
> A statistical assessment of the quality of our annotations is given in Sec. 4 of the paper. Since our method produces a binary confidence score for our generated label results, we validate the confidence score with Precision-recall curves in Fig. 4. The quantitative assessment of the quality of annotations on an open-source dataset with groundtruth information is shown in Table 3,4. Moreover, we also assess the quality of annotations as a separate quantitative ablation study in Appendix Sec. A. For the released dataset, we manually verify the correctness of labels and confirm the accuracy of the "confidence" flag before releasing the dataset.
>
> &nbsp;
>
> **[Weaknesses 2(b)] More analyses of the pipeline may be necessary, e.g., what should be done when the semi-supervised annotation quality is not good enough? Does adding more manual annotation help? If so, how many additional annotations would be meaningful?**
>
> The labels retained as "inliers" by MBW are chosen based on a very strict threshold of PCKh check [35] which is a standardized metric to calculate 2D errors that are agnostic to subject sizes or image projections. Before releasing the dataset, we confirm whether the annotations generated as "inliers" are good enough by visually confirming each sample thereby supporting the notion of strict PCKh checks that are incorporated in the MBW pipeline. The downside to this approach is - if the semi-supervised annotation did not have enough data to learn and bootstrap, chances would be that MBW would always reject the majority of candidates, and hence the quantity of labeled data set would never increase. We address this labeling issue by conducting an ablation study in Appendix Sec. A and Sec. B.
>
> From Fig. 8, we can see that adding manual annotations over specific contiguous chunks of samples should be enough to address the above issue. Although we did not have a second set of manual annotations (active learning), we observe that since we propose a principled way to detect outliers, our pipeline could be readily used in the active learning domain where the proposed approach of iterations can be useful to dictate the next set of labeling in active learning and thereby increasing the set of "inliers". Thus, if we have a contiguous chunk of outliers in space and time for our captured video sequence (as shown in Fig. 8), MBW could find this chunk of outliers in a principled way and can exactly specify where we should add more manual annotations.
>
> &nbsp;
>
> **[Weaknesses 2(c)] From Fig 4(a), increasing manual annotation from 1\% to 2\% does not seem to significantly improve the outlier rejection.**
>
> Increasing the manual annotation from 1\% to 2\% by a naive uniform sampling manner does not seem to significantly improve the outlier rejection. This observation is in accordance with one of our claims that only a small amount of labeled data is required initially (~1\%) to learn an initial 3D neural shape prior. Although this initial neural prior is not perfect, it is just enough to accomplish the task of rejecting the most egregious outliers. Since the NRSfM component exploits the 3D shape redundancy in geometry, naively increasing the samples from 1\% to 2\% does not add a substantial amount of information required for the task of outlier rejection.

---

> ### Author Response · Authors · 2022-08-14
> **Response to Reviewer QwEG feedback (Weaknesses #3 and Additional Feedback)**
>
> **[Weaknesses 3.] There is limited discussion on the "zoo dataset". It would be nice if the authors can provide more in-depth analyses of the dataset, e.g., what is the potential usage of the "zoo dataset" given it is annotated by the semi-supervised pipeline?**
>
> Although we state the detailed discussions on the “zoo dataset” in the attached supplementary as well as in the datasheet for the dataset: [MBW-Datasheet-for-Dataset Link](https://github.com/mosamdabhi/MBW-Data/tree/main/Datasheet_for_Dataset#uses), we restate the potential usage of the "zoo dataset" given the already provided labels:
>
> - **Computer Vision (Annotations for Keypoint prediction tasks):** Given that each frame consists of an associated "confidence" flag, the label frames with the "True" confidence flag could be used to train the 2D and 3D keypoint prediction models for articulated tail-end distribution categories.
>
> - **Computer Vision (Annotations for Action recognition tasks):** Given that we can generate bounding boxes (see Appendix Sec. D) for different object categories, we could potentially calculate the bounding box crops for different categories and use a heuristic that if certain bounding boxes overlap each other, there exists a certain action being conducted by the captured objects and thereby get action recognition labels for computer vision applications.
>
> - **Neuroscience experiments:** The 3D information of the tail-end distribution object categories such as animals, birds, and fish species could be used to understand their behaviors using the relative 3D structural information generated by MBW (with minimal human effort).
>
> &nbsp;
>
> **[Additional feedback 1.] More analyses on the robustness of the semi-supervised annotation pipeline would be helpful. For example, does it always work when the camera baseline is sufficiently large (all keypoints are visible from at least one view)? If not, what would be the typical failure pattern?**
>
> We observe that as long as a neural shape prior over varied poses is learned which could classify Out-of-Distribution ("outliers") frames, the iterative semi-supervised approach is feasible to generate "inliers" in the subsequent iterations. However, the typical failure pattern could arise from the following scenarios:
>
> - If the initial amount of labels is extremely small, resulting in a failure pattern similar to "mode-collapse", where the neural prior is unable to learn a varied pattern of non-rigid deformations.
> - Another failure pattern would arise when the baseline between cameras is extremely small rendering the neural prior unable to learn the shape variations via multi-view equivariance.
>
> &nbsp;
>
> **[Additional feedback 2.] The authors can showcase the value of the "zoo dataset" by applying it to some downstream tasks.**
>
> Although we show the value of the "zoo dataset" for the keypoint prediction task in the manuscript, we experiment to showcase the effects of this dataset for another computer vision downstream task -- generating labels for the action recognition task, and we plan to include this experiment as a downstream application example in the Appendix section. For example, if we want to recognize instances when the primate (Chimpanzee or Colobus Monkey) does certain actions such as bringing the hands closer to its mouth, we could easily generate labels for such actions using MBW. In this experiment, we generate labels to detect instances when the face bounding box overlaps with the hand bounding box. In this way, we annotate the frames where the specific action of hand-in-face-overlap (HIFO) is happening and then train a simple standalone HIFO classifier using the generated action recognition annotations. We measure the accuracy of this task using Precision-Recall(PR) AUC similar to Fig. 4(a,b) in the main paper -- in that, if the face is being overlapped by hands, it is considered as an "inlier" label and if the face is not overlapped, it is called an "outlier" for this action recognition task. Based on the above metric, we get the action recognition annotation PR AUC of **96\%**.

---

### Official Review · Reviewer_sAZo · 2022-07-27
**MBW**

**Rating:** 6
**Confidence:** 2
**Correctness:** Yes.

**Strengths:**

The ability to do achieve these results without calibration is worth highlighting. Calibration is a critical process and in many cases, essential. but by eliminating the need to calibrate, the proposed idea can expand to variety of applications.
To capture the 3D reconstructions of the unoccluded objects of interest with the minimal cameras is also worth mentioning.
The significantly low number of annotated frames needed for this method to work is impressive.
The comparison in Table 1 is also highly appreciated as a reader.
The approach is clearly laid out.
The paper is well written and easy to read.


**Weaknesses:**

The dataset being released is very small. It captures the essence of the proposed idea but seems lacking. Releasing a larger dataset with multitude of variations in addition to the mentioned ones would help further help (also to highlight the strength of the technique)
Impact of the technique on semi-occluded objects in the scene would further help.
In Appendix D, the idea for bounding box estimation and reducing the problem to single object-single frame sub problem is good but better techniques can be used for the same. This could help in reducing the number of iterations to solve this.
Also it would help to understand the impact of the bounding boxes not completely covering the Chimpanzee in the scene.


**Additional Feedback:**

Some shared in weaknesses.
What percentage of frames annotation had to be fixed by manual intervention? Which video had a higher percentage and any insights into the reasons for the same?
Wishing the very best to the authors and thank you for your contributions.

**Clarity:**

Yes. Especially appreciated was the Experiments section. Setting up the questions before jumping into the results is helpful, and very structured. There are some grammatical errors in the paper but easy to fix. For example, Fig 5(b) title - ".. denoise the improve the.."

**Documentation:**

Yes.

**Ethics:**

No.

**Relation To Prior Work:**

Yes

**Summary And Contributions:**

In this paper, the authors have combined a non-rigid 3D neural prior with deep flow to obtain high-fidelity landmark estimates from videos with only two or three uncalibrated, handheld cameras.  Utilizing Multi-View Non-Rigid Structure from Motion (MV-NRSfM) to more reliably estimate camera poses and 3D landmark positions from noisy 2D inputs with few cameras and leveraging deep optical flow, the paper presents a novel process. The dataset collected for this project is another contribution to the community.

---

> ### Author Response · Authors · 2022-08-14
> **Response to Reviewer sAZo feedback**
>
> **[Weaknesses 1.]: The dataset being released is very small. It captures the essence of the proposed idea but seems lacking. Releasing a larger dataset with multitude of variations in addition to the mentioned ones would help further help (also to highlight the strength of the technique) Impact of the technique on semi-occluded objects in the scene would further help.**
>
> We thank the reviewer for the valuable suggestion. We agree with your suggestion and plan to release the updated versions with more variations of the species as well as more versions of existing species where we plan to leverage the already existing "inlier" labels. In doing so, we propose to release a version of MBW with even fewer labels, sometimes as less as a single label for the first frame thereby making our approach "one-shot". The reasoning behind "one-shot" being: If we capture the data of similar species in a similar environment we can leverage the information learned earlier and hence go away with an even fewer amount ("one-shot") of data.
>
> &nbsp;
>
> **[Weaknesses 2]: In Appendix D, the idea for bounding box estimation and reducing the problem to single object-single frame sub problem is good but better techniques can be used for the same. This could help in reducing the number of iterations to solve this. Also it would help to understand the impact of the bounding boxes not completely covering the Chimpanzee in the scene.**
>
> We agree with the reviewer that more sophisticated bounding box estimation or object tracking techniques should be beneficial and we would appreciate it if the reviewer could point out any suggestion that could act as a replacement for the bounding box calculation technique we proposed in MBW.
>
> That being said, we find our current approach for bounding box calculation is not a bottleneck for the data labeling process, at least in the different scenarios we evaluated. For example, even if the bounding box does not completely cover the chimpanzee in the first iteration, this is not an issue since we calculate the bounding box over the reprojected 2D keypoints from MV-NRSfM, and also pad the bounding box crop image patch for the first iteration. In the subsequent iterations, the 2D keypoints and bounding box crops get more refined and we remove the need to pad the bounding box crop image patch as the MBW iterations progress. In extreme cases where the 2D keypoints are egregiously bad resulting in completely incorrect bounding box calculation, we simply drop the frame and request manual annotation for this frame, via the "Active learning" process mentioned in Appendix Sec. B.

---

### Official Review · Reviewer_5d5r · 2022-07-27
**A multi-view zoo dataset with a semi-supervised method**

**Rating:** 6
**Confidence:** 4
**Clarity:** Yes, the paper is overall well writte…

**Strengths:**

[S1] The paper create a zoo dataset with uncalibrated two synchronized views. The mutli-view animal datasets are rare and will be useful for future research.

[S2] The paper proposes a semi-supervised framework by leveraging MV-NRSfM and optical flow-based algorithms. MV-NRSfM handles the uncalibrated few views, while optical flow-based methods employ limited annotations to generate candidates. The whole framework run iteratively to perform auto-labeling.

[S3] The experiments show that they achieve similar performance with 1% annotations to fully supervised methods. Besides, the authors ablate the proposed method on the proposed dataset systematically


**Weaknesses:**

[W1] Technical novelty is limited. The merits of handling few uncalibrated views are originated from MV-NRSfM, and the 2d landmarks are generated from off-the-shelf flow-based model. The main techinical contribution is to use them in a semi-supervised scheme.

[W2] As shown in Fig. 4(a), the performance of MBW drops as the number of views increases. What's the reason behind that?

[W3] The experiments show that the method works well when there is only 1% annotation. What’s the minimum percentage for the algorithm to work well?

[W4] What will happen if optical flow-based model performs not very well? Any requirement for the minimal performance of optical flow-based model?

**Additional Feedback:**

NA

**Correctness:**

Some claims might be too strong, e.g., in abstract, the paper claim that "With just a few annotations ... obtain 3D reconstructions that are impossible with other existing approaches"

**Documentation:**

NA

**Ethics:**

No ethical concerns

**Relation To Prior Work:**

The authors clearly discussed how the proposed method and dataset differ from previous works

**Summary And Contributions:**

MBW are proposed to tackle two problems: (1) uncalibrated but synchronized videos with few views, and (2) a limited manual annotations. The main contributions lie in 3 aspects:
1. Propose a zoo dataset with uncalibrated two synchronized views
2. Propose a semi-supervised framework by leveraging MV-NRSfM and optical flow-based algorithms
3. Study different factors of the proposed method on the proposed dataset systematically

---

> ### Author Response · Authors · 2022-08-14
> **Response to Reviewer 5d5r feedback**
>
> **[W2]: As shown in Fig. 4(a), the performance of MBW drops as the number of views increases. What's the reason behind that?**
>
> This confusion arises from the fact that the Precision-Recall (PR) AUC bars per each view were calculated using the frames available in that view. Thus, a lower number of views corresponds to a lower number of frames. Thus, the PR AUC is calculated on a larger number of frames for the higher number of views. Although we make this distinction in Fig. 4 caption, we plan to make this abundantly clear in the final version to remove any confusion.
>
> &nbsp;
>
> **[W3]: The experiments show that the method works well when there is only 1\% annotation. What’s the minimum percentage for the algorithm to work well?**
>
> As long as the initial annotations are capturing geometrical poses with slight variations which could be considered valid for neural shape prior to rejecting and accepting the outliers, we confirm in our experiments, that even with 4-5 frames per view (which in some cases could be less than 1\% of the frames) MBW can generate >94\% correct predictions (certified by Eq. 2). We conduct the above experiment on an internal hands dataset and plan to release this study as part of the appendix in the supplementary section.
>
> &nbsp;
>
> **[W4]: What will happen if the optical flow-based model performs not very well? Any requirement for the minimal performance of the optical flow-based model?**
>
> The minimal requirement for the optical flow-based model to work is that the frames should be sequential. If we do not have access to the sequential data, resulting in an optical-flow-based model not performing very well, this should result in the neural shape prior rejecting all the samples from the optical flow — provided that the neural prior has learned a reasonable shape manifold to reject the outliers from the optical flow based model. In such a scenario, we can discard the optical flow model iteration and only execute the iterations with MV-NRSfM and 2D Detector. We can do the above since MV-NRSfM can learn a neural shape prior even with a very small amount (1 \% of frames per view) capable of rejecting the outliers. However, doing so would only accept the geometrical poses that it has roughly seen before and reject all the pose variations that it has not seen. If MV-NRSfM is unable to add any 2D label predictions as inliers after multiple iterations, we can then actively label a chunk of frames by analyzing the output such as Fig. 8 in the Appendix section, which could help us dictate the next set of labeling, which should correspond to the best possibility of increasing the inliers.
>
> &nbsp;
>
> **[Correctness]: Some claims might be too strong, e.g., in the abstract, the paper claim that "With just a few annotations ... obtain 3D reconstructions that are impossible with other existing approaches".**
>
> To the best of our knowledge, no other approach exists that enables the generation of 3D structures of objects captured in the wild without any calibration. That being said, we would tone down the claim and clarify the above reasoning behind this claim in the final version.

---

### Author Response · Authors · 2022-08-14
**General response to the reviewers.**

We thank the reviewers for their constructive feedback and appreciating the fact that MBW could be a powerful way of generating labeled data in the wild at scale — thereby democratizing the domain of data collection to a plethora of machine learning-driven applications. Moreover, we also thank the reviewers for noting that the paper is well-written, clearly motivated, relation to prior work is duly cited, and the released dataset is well-documented.

We answer the concern of novelty raised by Reviewer 5d5r and Reviewer QwEG and then address the specific concerns in the individual comments to each reviewer.

&nbsp;
### Technical novelty

**Data labeling approach:** We propose a scalable data collection approach that leverages the advances in Non-rigid Structure-from-Motion (NRSfM) and machine learning. The use of techniques from NRSfM to date has not found utility in the domain of semi-supervision literature, which primarily uses techniques from SfM [1,13,30]. In this paper, we demonstrate a novel approach that incorporates the techniques from NRSfM [2] -- giving us an automated way to determine the inliers to be used in a self-training or bootstrapping fashion _for_ data labeling. In this work, we observe that the advantage of NRSfM comes from robustly detecting outliers and denoising the inlier 2D landmark estimates by leveraging a learned 3D shape manifold. Finally, we propose to combine the spatial constraints from NRSfM with temporal constraints from an optical flow approach that helps the proposed semi-supervision process, especially in the initial iterations.


**In-the-wild dataset:** We validate the proposed approach by releasing a labeled dataset that is collected in an unconstrained in-the-wild setup with two uncalibrated handheld camera views. We show that our approach can generate labels (along with a confidence flag) with an effort of a mere few minutes. We believe that MBW could be used by the machine learning and computer vision community to generate in-the-wild datasets at scale by using the methodologies proposed in the paper as well as using the corresponding codebase that is scheduled to be an open-source release to the public.

---

### Meta-Review · Area_Chair_NtYv · 2022-09-11

**Recommendation:** Accept
**Confidence:** 4

**Metareview:**

This paper presents a dataset of uncalibrated pairs/triplets of videos of zoo animals. The paper also presents a method to label this data in a self-supervised manner with minimal annotation. Reviewers appreciated the contents of this dataset and see value for follow up work. The method for labelling was also appreciated. Weaknesses of the paper have been addressed or acknowledged. I recommend accpeting this paper to the NeurIPS 2022 Datasets and Benchmarks program.

---

### Decision · Program_Chairs · 2022-09-16

Accept